# OpenReview forum: "ReFIne: A Framework for Trustworthy Large Reasoning Models with Reliability, Faithfulness, and Interpretability"
_ICLR.cc/2026/Conference — Submitted to ICLR 2026_

### Official Review · Reviewer_Npia · 2025-10-30

**Soundness:** 2
**Presentation:** 3
**Contribution:** 2
**Rating:** 4
**Confidence:** 3

**Summary:**

The paper proposes ReFIne, a two-stage training framework (SFT + GRPO) designed to improve Large Reasoning Models along three dimensions: interpretability, faithfulness, and reliability. The authors separate reasoning into distinct phases (understanding, facts, plan, think, self-assessment).  They also perform RL with specifically designed rewards to enhance such behavior. They evaluate ReFIne on Qwen3 models (1.7B–8B) across four mathematical benchmarks, reporting substantial improvements in all three trustworthiness metrics while maintaining competitive accuracy and even improving token efficiency.

**Strengths:**

a. The introduction of interpretability, faithfulness, and reliability is well-motivated and addresses real gaps in language model evaluation.

b. The paper includes multiple quantitative metrics for quantitative evaluation of trustworthy language reasoning models such as cross-section referencing rates, disclosure faithfulness.

c. RL with rewards designed for cross-section reference and confidence estimation can largely boost the performance in interpretability, faithfulness and reliability.

d. The structured reasoning leads to shorter reasoning traces with competitive accuracy with Plain-Qwen in this study.

**Weaknesses:**

a.Readability and faithfulness are judged by QwQ-32B, no human evaluation is reported in this study.

b. The Plain-Qwen model is a weak baseline for comparison, as it is only trained on the thinking processes of the SFT dataset. This does not provide a robust comparison against a strong, general-purpose CoT or reasoning model. The paper should ideally report the performance of a more capable model, such as Qwen3-8B-Thinking, on the novel metrics (interpretability, faithfulness, and reliability) to provide a more meaningful context.

c. There appears to be a discrepancy between the reported results and the qualitative evidence. Specifically, Figure 4 suggests that ReFine achieves shorter reasoning traces, yet the examples provided in the Appendix demonstrate traces that are noticeably much longer than those generated by Plain-Qwen. This contradiction requires clarification or re-evaluation.

d. The proposed trustworthiness metrics appear highly coupled with the specific templates and reasoning structure used in this study. It is unclear if these metrics generalize to models trained with different reasoning formats, potentially limiting their broader applicability in the field.

**Questions:**

See Weaknesses

---

> ### Author Response · Authors · 2025-11-28
> **Rebuttal (1/3)**
>
> Dear reviewer Npia,
>
> Thank you for your valuable feedback! We address your questions and concerns below.
>
> > **Q1:** Readability and faithfulness are judged by QwQ-32B, no human evaluation is reported in this study.
>
> **A1:** Thank you for the suggestion. We agree that human evaluation is valuable when it provides a meaningful signal. However, for readability, a human study would not be informative in our setting: ReFIne outputs clearly separated and labeled sections, whereas Plain-Qwen3 produces a single unstructured CoT. Human annotators can distinguish these formats instantly, making the task a test of format detectability rather than an evaluation of genuine readability quality. For this reason, we did not conduct a separate human study on interpretability.
>
> For faithfulness, human judgment is meaningful, so we conducted an additional human evaluation mirroring the commitment-faithfulness test in Section 4.2. We evaluated ReFIne-qwen3-8b on GSM8K by presenting 10 problems and complete reasoning traces to 4 annotators. Each annotator rated whether the \<think> reasoning was grounded in the \<understanding>, \<facts>, and \<plan> sections using a five-point scale:
>
> * 100% (Fully aligned)
> * 75% (Mostly aligned)
> * 50% (Partially aligned)
> * 25% (Weakly aligned)
> * 0% (Not aligned)
>
> The aggregated results are shown in Table A. Human evaluators found the model’s reasoning to be highly grounded: alignment with the stated understanding and facts is nearly perfect (98–100%), and alignment with the step-by-step plan remains strong (92%). These findings confirm that the commitment-faithfulness behavior measured by QwQ-32B in the main paper closely matches human-judged faithfulness.
>
> ### Table A: Commitment faithfulness. For each dataset, we report the fraction of traces where \<think> strictly follow \<understanding> / \<facts> / \<plan>
>
> | Model            | GSM8K             |
> |------------------|-------------------|
> | ReFIne-qwen3-8b  | 98% / 100% / 92%  |

---

> ### Author Response · Authors · 2025-11-28
> **Rebuttal (2/3)**
>
> > **Q2:** The Plain-Qwen model is a weak baseline for comparison, as it is only trained on the thinking processes of the SFT dataset. This does not provide a robust comparison against a strong, general-purpose CoT or reasoning model. The paper should ideally report the performance of a more capable model, such as Qwen3-8B-Thinking, on the novel metrics (interpretability, faithfulness, and reliability) to provide a more meaningful context.
>
> **A2:** Thank you for the comment! To clarify, we would like to note that the suggested comparison between Qwen3-8B-Thinking is actually not a fair comparison. The reason is that Qwen3-8B-Thinking is trained on a much larger and proprietary reasoning dataset that we do not have access to. Due to its training corpus and supervision signals being unknown and substantially richer than ours, comparing ReFIne directly against Qwen3-8B-Thinking would conflate data advantages with methodological differences and would not isolate the contribution of structured reasoning.
>
> In contrast, Plain-Qwen3 is trained under the exact same conditions as ReFIne, same SFT data, same GRPO data, and same compute budget. This makes Plain-Qwen3 the only controlled and methodologically meaningful baseline, as it isolates the effect of adding structure while holding all other factors constant.
>
> That said, to provide a broader context, we also report the interpretability, faithfulness, and reliability metrics for Qwen3-8B-Thinking in Tables B, C, and D below. Although its training data advantage makes this comparison inherently uneven, ReFIne still shows stronger performance across all three trustworthiness dimensions, demonstrating that structured reasoning can improve properties that standard CoT reasoning does not explicitly target.
>
> ### Table B: Comparison of interpretability (readability) judged by QwQ-32B
> | Size |  | ReFIne clearly better | ReFIne slightly better | Qwen3 slightly better | Qwen3 clearly better |
> |---|--|---|---|---|---|
> | 1.7B | ReFIne vs Qwen3 | 64.55% | 7.48% | 10.56% | 17.41% |
> | 4B   | ReFIne vs Qwen3 | 59.95% | 8.06% | 13.77% | 18.21% |
> | 8B   | ReFIne vs Qwen3 | 65.29% | 9.23% | 12.44% | 12.03% |
>
> ### Table C: Disclosure Faithfulness
>
> | Size | Model  | AIME2024 | GPQA-Diamond | MATH-500 | GSM8K |
> |---|---|---|---|---|---|
> | 1.7B | ReFIne | **0.733 ± 0.091** | 0.863 ± 0.025 | 0.829 ± 0.037 | 0.749 ± 0.038 |
> |  | Qwen3 | 0.690 ± 0.101 | **0.914 ± 0.025** | **0.844 ± 0.051** | **0.810 ± 0.025** |
> | 4B | ReFIne | **0.956 ± 0.064** | **0.910 ± 0.026** | **0.927 ± 0.043** | **0.983 ± 0.010** |
> |  | Qwen3 | 0.670 ± 0.167 | 0.830 ± 0.033 | 0.663 ± 0.076 | 0.826 ± 0.067 |
> | 8B | ReFIne | **0.957 ± 0.060** | **0.856 ± 0.039** | **0.934 ± 0.036** | **0.966 ± 0.024** |
> |  | Qwen3 | 0.617 ± 0.198 | 0.776 ± 0.049 | 0.649 ± 0.127 | 0.867 ± 0.036 |
>
> ### Table D: Reliability (Confidence verbalization rate ↑ / AUROC ↑ / ECE ↓)
> | Size | Model | AIME2024 | GPQA-Diamond | MATH-500 | GSM8K |
> |---|---|---|---|---|---|
> | 1.7B | ReFIne | **100.0%** / **0.795** / **0.305** | **99.4%** / **0.584** / **0.279** | **100.0%** / **0.726** / **0.080** | **100.0%** / **0.605** / **0.118** |
> |  | Qwen3 | 27.0% / 0.669 / 0.572 | 43.8% / 0.603 / 0.512 | 54.4% / 0.522 / 0.292 | 88.6% / 0.510 / 0.440 |
> | 4B   | ReFIne | **100.0%** / **0.872** / 0.204 | **99.6%** / 0.649 / **0.274** | **100.0%** / **0.757** / **0.042** | **100.0%** / **0.621** / **0.075** |
> |  | Qwen3 | 57.2% / 0.737 / **0.117** | 89.4% / **0.672** / 0.304 | 94.8% / 0.354 / 0.187 | 99.9% / 0.516 / 0.860 |
> | 8B   | ReFIne | **100.0%** / 0.763 / 0.179 | **99.8%** / 0.679 / **0.196** | **100.0%** / **0.713** / **0.032** | **100.0%** / **0.677** / **0.043** |
> |  | Qwen3 | 26.1% / **0.995** / **0.137** | 51.8% / **0.749** / 0.273 | 78.4% / 0.505 / 0.209 | 95.2% / 0.449 / 0.905 |
>
> ---
>
> > **Q3:** There appears to be a discrepancy between the reported results and the qualitative evidence. Specifically, Figure 4 suggests that ReFine achieves shorter reasoning traces, yet the examples provided in the Appendix demonstrate traces that are noticeably much longer than those generated by Plain-Qwen. This contradiction requires clarification or re-evaluation.
>
> **A3:** Thank you for pointing this out. To clarify, the examples in the Appendix are cropped illustrations, not full reasoning traces. We intentionally displayed more of the ReFIne outputs because the purpose of the appendix examples is to demonstrate the cross-section grounding behavior (i.e., how ReFIne cites <understanding>, <facts>, and <plan> throughout the reasoning). Showing only a short excerpt would not make this behavior visible, so we included a larger portion of the ReFIne trace.
>
> In contrast, the Plain-Qwen examples are shorter excerpts simply because there is no structured referencing behavior to demonstrate. As a result, the appendix examples may appear longer for ReFIne, even though the full trajectories are shorter.
>
> We will make this point clearer in the revision to avoid confusion.

---

> ### Author Response · Authors · 2025-11-28
> **Rebuttal (3/3)**
>
> > **Q4:** The proposed trustworthiness metrics appear highly coupled with the specific templates and reasoning structure used in this study. It is unclear if these metrics generalize to models trained with different reasoning formats, potentially limiting their broader applicability in the field.
>
> **A4:** Thank you for raising this point. To clarify, although our reasoning format uses explicit sections, the trustworthiness dimensions we evaluate are not specific to our template and can be applied to any reasoning model regardless of structure. More specifically,
>
> * **Reliability** is entirely template-agnostic. AUROC and ECE are standard metrics that measure whether a model’s self-reported confidence matches correctness. These apply equally to free-form CoT.
>
> * **Faithfulness** is also not tied to our tags. The two forms we evaluate, commitment faithfulness and disclosure faithfulness, are based on general causal criteria:
>   1. Does the model’s later reasoning follow from its earlier commitments?
>   2. Does the model acknowledge external cues (e.g., hints) when they influence its answer?
>
>   These principles were introduced independently of our framework and can be applied to any model with or without explicit section markers.
>
> * **Interpretability** in our study is measured through readability, which again does not require a specific template. Any model that outputs intermediate reasoning can be evaluated on whether the reasoning is coherent, organized, and human-traceable.
>
> In short, these metric definitions themselves are broad and widely applicable to diverse reasoning formats. ReFIne provides one instantiation of these trustworthiness criteria, but the metrics are not limited to our template and can be used to evaluate other CoT or reasoning-style models as well.
>
> ---
>
> # Summary
>
> In summary, we address the reviewer’s concerns on human evaluation, baseline fairness, appendix examples, and metric generality in **A1–A4**:
>
> 1. **In A1**, we explain why readability does not benefit from human evaluation but provide a new human study for faithfulness, showing near-perfect alignment between human judgments and QwQ-32B (98–100% for \<understanding>/\<facts> and 92% for \<plan>).
>
> 2. **In A2**, we clarify that Qwen3-8B-Thinking is not a fair baseline due to its proprietary training corpus and that Plain-Qwen3 is the only controlled comparison. We additionally report trustworthiness metrics for Qwen3-8B-Thinking and show that ReFIne still performs better across interpretability, faithfulness, and reliability.
>
> 3. **In A3**, we resolve the apparent discrepancy in trace length by explaining that Appendix examples are intentionally cropped to highlight cross-section referencing behavior, while actual full trajectories remain shorter for ReFIne.
>
> 4. **In A4**, we clarify that the trustworthiness metrics are not tied to our template and generalize broadly: reliability uses standard confidence calibration metrics, faithfulness relies on causal criteria, and readability evaluates reasoning coherence independent of format.
>
> We believe our response addresses all of the reviewer’s concerns. Please let us know if you have further questions; we would be happy to clarify and discuss further.

---

### Official Review · Reviewer_RVD7 · 2025-10-31

**Soundness:** 3
**Presentation:** 3
**Contribution:** 3
**Rating:** 6
**Confidence:** 4

**Summary:**

The paper focuses on the trustworthiness of LRMs.
The authors define trustworthiness along three dimensions: interpretability, faithfulness, and reliability.
They propose ReFIne, a two-stage training framework to instill these properties, including SFT and GRPO. Experiments on Qwen3 models across four mathematical benchmarks show that ReFIne significantly improves the trustworthiness while maintaining comparable accuracy and slightly improving token efficiency compared to baseline models.

**Strengths:**

1. The paper provides a clear, three-part definition for trustworthy reasoning, moving beyond the standard metrics of accuracy and efficiency. This framework addresses a critical and often-overlooked gap in LRM development.
2. The ReFIne framework is well-designed, combining SFT to teach the desired structure and GRPO to enforce the target behaviors.
3. The framework demonstrates significant quantitative improvements across all three target dimensions.

**Weaknesses:**

1. Several key metrics rely on another large model for evaluation. This introduces potential evaluation bias, and the results would be stronger if supplemented with human evaluation to confirm that the model-judged improvements translate to a better human-user experience.
2. While generally comparable, the ReFIne models show a slight decrease in accuracy on the AIME-2024. Although minor, this suggests a potential trade-off between enforcing a strict, trustworthy structure and achieving peak accuracy on highly complex problems.

**Questions:**

1. The ReFIne framework proves highly effective for structured, step-by-step mathematical problems. How could this framework be adapted for tasks that are inherently less structured, such as commonsense reasoning, where some of the commonsense cannot be verified?

---

> ### Author Response · Authors · 2025-11-28
> **Rebuttal (1/2)**
>
> Dear reviewer RVD7,
>
> Thank you for your valuable feedback! In the rebuttal below we would like to answer your questions and address your concerns.
>
> > **Q1:** The results would be stronger if supplemented with human evaluation to confirm that the model-judged improvements translate to a better human-user experience.
>
> **A1:** Thank you for the suggestion! We conducted an additional human study to verify whether the QwQ-based judgments used in our paper align with human perception. Specifically, we evaluated ReFIne-qwen3-8b on GSM8K, presenting 10 problems and their full reasoning traces to 4 human annotators. For each example, annotators assessed whether the model’s \<think> reasoning was grounded in the \<understanding>, \<facts>, and \<plan> sections, following the same commitment-faithfulness protocol as Section 4.2, but using a five-point scale instead of a binary decision:
> * 100% (Fully aligned)
> * 75% (Mostly aligned)
> * 50% (Partially aligned)
> * 25% (Weakly aligned)
> * 0% (Not aligned)
>
> The aggregated human ratings are summarized in the table below. The results show that human evaluators found the model’s reasoning to be highly grounded with the <understanding>, <tact>, <plan> sections. In particular, alignment with the stated understanding and factual information is nearly perfect (98–100%), and even alignment with the detailed step-by-step plan remains very strong (92%). This confirms that the commitment-faithfulness behavior measured by QwQ in the main paper closely matches human-judged faithfulness.
>
> ### Table A: Commitment faithfulness. For each dataset, we report the fraction of traces where \<think> strictly follow \<understanding> / \<facts> / \<plan>
>
> | Model              | GSM8K                |
> |--------------------|----------------------|
> | ReFIne-qwen3-8b    | 98% / 100% / 92%     |
>
> ---
>
> > **Q2:** While generally comparable, the ReFIne models show a slight decrease in accuracy on the AIME-2024. Although minor, this suggests a potential trade-off between enforcing a strict, trustworthy structure and achieving peak accuracy on highly complex problems.
>
> **A2:** We appreciate the reviewer’s insightful observation. We’d like to note that while ReFIne does show a small decrease on AIME-2024, this pattern is not consistent across benchmarks. In particular, ReFIne actually achieves higher accuracy on GPQA, a dataset that is generally considered more challenging than AIME, indicating that structured reasoning is not inherently detrimental and might help stabilize reasoning in complex scientific domains.
>
> More importantly, ReFIne is a multi-objective framework designed to improve interpretability, faithfulness, reliability, and reasoning consistency. These dimensions are often in mild tension with raw accuracy, and such tradeoffs are well-documented in prior work (e.g., alignment and trustworthiness interventions frequently reduce peak reasoning performance [1]). In our results, this tradeoff is modest, while the gains in trustworthiness are substantial.
>
> We believe this reflects the broader reality that building trustworthy LRMs requires balancing accuracy with additional properties rather than optimizing a single metric.
>
> [1] Huang, Y., Wang, S., Chen, T., et al. Safety Tax: Safety Alignment Makes Your Large Reasoning Models Less Reasonable. 2025.

---

> ### Author Response · Authors · 2025-11-28
> **Rebuttal (2/2)**
>
> > **Q3:** How could this framework be adapted for tasks that are inherently less structured, such as commonsense reasoning, where some of the commonsense cannot be verified?
>
> **A3:** Thank you for the insightful question. Commonsense reasoning differs fundamentally from mathematics problem-solving because many intermediate steps cannot be objectively verified. To examine whether ReFIne remains beneficial under this setting, we additionally evaluated it on CommonsenseQA. As shown in Table B, ReFIne’s accuracy is slightly lower than the Plain models. This modest accuracy drop is expected: ReFIne jointly optimizes multiple objectives (interpretability, faithfulness, reliability, and accuracy), and balancing these goals can produce small accuracy shifts depending on the task.
>
> Despite this difference, ReFIne consistently improves the key trustworthiness properties. QwQ judge found ReFIne’s reasoning to be substantially more readable across model sizes (Table C), indicating that the structured reasoning framework still organizes the thought process more clearly even when the domain is less formally structured. ReFIne also achieves significantly higher disclosure faithfulness (Table D), meaning the model is more likely to reveal its true underlying thinking process. This demonstrates that the framework continues to regularize and discipline the reasoning process in a beneficial way, even when the task does not supply rigidly checkable intermediate steps.
>
> Regarding reliability (Table E), although the AUROC and ECE values of the Plain model are numerically better in some cases, this comparison is affected by differences in confidence verbalization behavior. Plain models do not consistently verbalize confidence. When reliability metrics are computed over only a fraction of the data, the resulting AUROC/ECE values may not reflect the model’s overall uncertainty calibration. ReFIne, in contrast, verbalizes confidence in nearly all examples across all model sizes, providing full-coverage reliability signals that are more aligned with downstream use.
>
> Overall, while commonsense reasoning contains inherent uncertainty that makes perfect verification impossible, ReFIne still improves interpretability, faithfulness, and reliability, while maintaining competitive accuracy.
>
> ### Table B: Accuracy comparison on CommonsenseQA
>
> | Size | Model   | CommonsenseQA    |
> |---|----|---|
> | 1.7B | ReFIne  | 66.99% ± 0.93%  |
> |      | Plain   | 71.90% ± 0.74%    |
> | 4B   | ReFIne  | 76.63% ± 0.84%    |
> |      | Plain   | 79.15% ± 0.26%    |
> | 8B   | ReFIne  | 80.36% ± 0.61%   |
> |      | Plain   | 83.05% ± 0.65%   |
>
> ### Table C: Readability comparison on CommonsenseQA
>
> | Size |  | ReFIne clearly better | ReFIne slightly better | Plain slightly better | Plain clearly better |
> |---|--|---|---|-----|---|
> | 1.7B | ReFIne vs Plain | 73.13% | 8.65% | 10.45% | 7.77% |
> | 4B   | ReFIne vs Plain | 74.33% | 7.46% | 10.97% | 7.24% |
> | 8B   | ReFIne vs Plain | 65.57% | 11.04% | 15.73% | 7.66% |
>
> ### Table D: Disclosure faithfulness on CommonsenseQA
>
> | Size | Model    | CommonsenseQA     |
> | --- | --- | --- |
> | 1.7B | ReFIne | **0.938 ± 0.014** |
> |      | Plain      | 0.663 ± 0.014     |
> | 4B   | ReFIne | **0.958 ± 0.010** |
> |      | Plain      | 0.848 ± 0.021     |
> | 8B   | ReFIne     | **0.966 ± 0.006** |
> |      | Plain      | **0.966 ± 0.009**     |
>
> ### Table E: Reliability (Confidence verbalization rate ↑ / AUROC ↑ / ECE ↓) on CommonsenseQA
>
> | Size | Model      | AIME2025    |
> | --- | --- | --- |
> | 1.7B | ReFIne | **100.0%** / **0.606** / **0.185** |
> |      | Plain      | 18.2% / 0.575 / 0.281   |
> | 4B   | ReFIne | **99.9%** / 0.663 / 0.214   |
> |      | Plain  | 91.0% / **0.704** / **0.043**  |
> | 8B   | ReFIne | **100.0%** / 0.643 / 0.227     |
> |      | Plain      | 93.9% / **0.695** / **0.068**    |
> ---
>
> # Summary
>
> In summary, we address the reviewer’s concerns on human evaluation, accuracy tradeoffs, and adaptation to less structured tasks in **A1–A3**:
>
> 1. **In A1**, we conduct an additional human study confirming that human annotators overwhelmingly agree with QwQ’s commitment-faithfulness judgments, showing 98–100% alignment for \<understanding> and \<facts>, and 92% for \<plan>.
>
> 2. **In A2**, we clarify the modest nature of the AIME accuracy drop, highlight that ReFIne improves accuracy on GPQA (a harder benchmark), and explain that mild accuracy–trustworthiness tradeoffs are well-documented and expected in multi-objective alignment settings.
>
> 3. **In A3**, we extend evaluation to CommonsenseQA and show that although accuracy is slightly lower, ReFIne consistently improves readability, disclosure faithfulness, and reliability across all model sizes, demonstrating that structured reasoning remains beneficial even for tasks without strictly verifiable intermediate steps.
>
> We believe our response addresses all of the reviewer’s concerns. Please let us know if you have further questions; we would be happy to clarify and discuss further.

---

### Official Review · Reviewer_BTMG · 2025-10-31

**Soundness:** 3
**Presentation:** 3
**Contribution:** 3
**Rating:** 4
**Confidence:** 4

**Summary:**

This paper presents ReFIne training framework that aims to enhance the trustworthiness of large reasoning models from three dimensions: interpretability, faithfulness, and reliability. ReFIne integrates supervised fine-tuning with GRPO optimization and adopts a structured, tag-based reasoning format (e.g., understanding, facts, plan, thinking, final answer, and self-assessment). It further incorporates cross-section references to highlight decisive information and generates calibrated confidence estimates for model outputs. The ReFIne is evaluated on reasoning models with different parameter scales using mathematical benchmarks of varying difficulty. Experimental results show that ReFIne improves the clarity of reasoning traces, the accuracy of causal attribution, and the calibration of confidence estimates, while maintaining accuracy and modestly improving reasoning efficiency.

**Strengths:**

1. The paper addresses the important challenge of enhancing the trustworthiness of reasoning models through a structured training framework that jointly optimizes interpretability, faithfulness, and reliability.

2. The integration of supervised fine-tuning and GRPO within the structured tag-based reasoning format is coherent and purpose-driven, effectively aligning training objectives with model trustworthiness.

3. The experimental results show improvements across multiple evaluation metrics, indicating the effectiveness of the proposed framework.

**Weaknesses:**

1. The distinction between ReFIne and existing reasoning training paradigms is not clearly articulated.

2. The proposed definitions of interpretability, faithfulness, and reliability lack theoretical justification and formal grounding.

3. The paper lacks ablation and sensitivity analyses to verify the contribution and stability of each component.

4. The experiments are limited to mathematical reasoning tasks, which restricts the generalization of the framework.

5. The paper does not analyze computational cost or the trade-off between efficiency and performance.

**Questions:**

1. The paper should provide a more thorough discussion of existing reasoning training paradigms and clearly distinguish them from the proposed ReFIne framework. The authors are advised to identify the remaining challenges in prior work and explicitly show how ReFIne addresses these gaps to better emphasize its novelty and contribution.

2. The paper introduces new definitions of interpretability, faithfulness, and reliability as dimensions of model trustworthiness and proposes corresponding metrics for their evaluation. However, these concepts currently lack theoretical justification or formal grounding, which undermines their credibility. The authors are encouraged to include theoretical analysis or cite relevant studies to strengthen the conceptual foundation of these definitions.

3. The paper should include ablation experiments on reward weighting, confidence calibration, and reference-linking incentives to verify the contribution of each component. It is also advisable to analyze the sensitivity of the framework to random seeds and data order to ensure that the reported improvements stem from specific design choices rather than training instability. In addition, the authors should test ReFIne under conditions with noisy or missing verification cues to examine whether it maintains reliable attribution and calibrated confidence in more challenging scenarios.

4. The generalizability of the proposed framework is central to its practical value. However, the experiments are confined to mathematical reasoning tasks, which limits the evidence for broader applicability. The authors should extend the evaluation to other reasoning domains, such as commonsense reasoning, scientific question answering, and code analysis, and assess performance on additional model families beyond the Qwen series.

5. The paper should include a concise analysis of computational cost and efficiency by reporting token overhead, reasoning latency, and memory usage. It is also recommended to investigate whether ReFIne can retain comparable improvements when using fewer or simplified reasoning sections, thereby clarifying the balance between efficiency and effectiveness.

**Details Of Ethics Concerns:**

No ethics concerns are identified.

---

> ### Author Response · Authors · 2025-12-03
> **Rebuttal (1/6)**
>
> Dear reviewer BTMG,
>
> Thank you for your valuable feedback! In the rebuttal below we would like to answer your questions and address your concerns.
>
> > **Q1:** The paper should provide a more thorough discussion of existing reasoning training paradigms and clearly distinguish them from the proposed ReFIne framework. The authors are advised to identify the remaining challenges in prior work and explicitly show how ReFIne addresses these gaps to better emphasize its novelty and contribution.
>
> **A1:** Thank you for the suggestion. We discuss existing reasoning training paradigms and distinguish them from our work below.
>
> Existing reasoning-model training pipelines mainly optimize correctness using RL (e.g., DeepSeek-R1 applies GRPO with a correctness reward to build stronger reasoning capabilities). These methods do not restructure the overall reasoning process, nor do they explicitly target interpretability, faithfulness, or reliability.
>
> In contrast, ReFIne introduces a multi-phase structured reasoning format and directly reinforces cross-section grounding and confidence-aware self-assessment, addressing three major limitations of prior work:
> (i) reasoning traces that are difficult to follow or unnecessarily long [1];
> (ii) weak faithfulness, for example, the model failing to honestly acknowledge external cues such as hints when they influence the solution [2];
> (iii) poor confidence calibration [3].
>
> Together, these components position ReFIne as a complementary framework that moves beyond accuracy-centric training and improves the broader trustworthiness of reasoning models.
>
> [1] Sui, Y., et al. Stop Overthinking: A Survey on Efficient Reasoning for Large Language Models. 2025.
> [2] Chen, A., et al. Reasoning Models Don’t Always Say What They Think. Anthropic, 2025.
> [3] Mei, X., et al. Reasoning About Uncertainty: Do Reasoning Models Know When They Don’t Know? 2025.
>
> > **Q2:** The paper introduces new definitions of interpretability, faithfulness, and reliability as dimensions of model trustworthiness and proposes corresponding metrics for their evaluation. However, these concepts currently lack theoretical justification or formal grounding, which undermines their credibility. The authors are encouraged to include theoretical analysis or cite relevant studies to strengthen the conceptual foundation of these definitions.
>
> **A2:** We agree that purely axiomatic definitions of interpretability, faithfulness, and reliability are still an open research problem. Our goal in this work is therefore more operational: to make these dimensions precise enough to measure in LRMs, and to ground them in existing literature rather than define them in a vacuum.
>
> More specifically, we clarify each metric for the reviewer’s reference below:
> For Reliability, this metric is actually already formally grounded: our definition corresponds directly to probabilistic calibration (the model “knowing when it knows”), and we measure it using standard, theoretically motivated calibration metrics such as AUROC and ECE, which are widely used in uncertainty and calibration studies for neural networks and LLMs.
>
> For faithfulness, we explicitly build on the disclosure faithfulness paradigm introduced by Reasoning Models Don’t Always Say What They Think (Chen et al., 2025), where a hint is injected into the prompt and one checks whether cases that flip from incorrect→correct explicitly acknowledge using that hint. Our “disclosure faithfulness” metric is a direct adaptation of this now widely used setup, and we additionally introduce commitment faithfulness to test whether the \<think> derivation adheres to the model’s own \<understanding>, \<facts>, and \<plan> sections.
>
> For interpretability, we focus on structural readability and organization – properties that are inherently more subjective but still central in the broader interpretability literature. We operationalize this via (i) explicit structural constraints (tag ordering, cross-section references) and (ii) readability judgments from QwQ-32B as an automatic proxy for human evaluation, following recent practice of using strong models as judges at scale.
>
> We will clarify in the revision that our contribution is to instantiate these trustworthiness dimensions in a concrete, measurable way for LRMs, grounded in existing work on calibration and reasoning faithfulness, rather than to provide a full formal theory. We believe this empirical grounding and alignment with prior paradigms substantially strengthen the credibility of our definitions and metrics.

---

> ### Author Response · Authors · 2025-12-03
> **Rebuttal (2/6)**
>
> > **Q3:** The paper should include ablation experiments on reward weighting, confidence calibration, and reference-linking incentives to verify the contribution of each component.
>
> **A3:** We agree that understanding the contribution of each reward component is important. However, conducting full ablation studies across all model sizes is computationally expensive, a single GRPO run requires roughly two days, making large-scale ablations infeasible. To keep the study tractable, we ablate the two key reward terms (cross-section reference reward and confidence-calibration reward) on the smallest model (1.7B). Although this setting is inherently more variable and noisier, it still provides useful signal regarding the impact of each reward component.
>
> Across all metrics, accuracy, interpretability, disclosure faithfulness, and reliability, the full ReFIne model consistently outperforms both ablated variants:
> Accuracy (Table A): Removing either the cross-section reference reward or the confidence-calibration reward leads to drops on most datasets. The full model maintains the strongest and most stable performance across all the benchmarks.
>
> Interpretability (Table B): QwQ-32B ratings show that ReFIne produces more readable and structured reasoning compared to the base model. Both ablated variants reduce interpretability benefits, confirming that the reward terms contribute meaningfully to reasoning clarity.
>
> Disclosure Faithfulness (Table C): The full ReFIne achieves the highest faithfulness on most datasets. Removing the cross-section reference reward notably harms faithfulness, especially on AIME2024 and GPQA, highlighting its importance for consistent, well-referenced reasoning traces.
>
> Reliability (Table D): All variants maintain high confidence verbalization, but the full model shows the most stable behavior across datasets. Removing the cross-section reference reward consistently reduces AUROC and increases ECE, confirming its importance for reliable confidence signals. The confidence-calibration ablation shows mixed effects, occasionally higher AUROC but worse ECE, indicating it does not provide a clear standalone benefit, though it also does not harm performance. Overall, the full model achieves the most balanced and stable reliability profile.
>
> Overall, while the small-model ablations are noisy, the trend is clear: when considering all metrics, accuracy, interpretability, faithfulness, and reliability, the full ReFIne model consistently performs better than either ablation.
>
> ### Table A: Accuracy comparison with reward ablations
> | Size | Model | AIME2024 | GPQA-Diamond | MATH-500  | GSM8K  |
> |---|---|----|---|---|---|
> | 1.7B | ReFIne | 29.33% ± 4.10%      | 25.45% ± 1.97%      | 83.78% ± 1.19%      | 84.09% ± 0.63%     |
> | | ReFIne w/o cross-section reference reward | 28.00% ± 5.26%   | 22.27% ± 2.07%      | 81.84% ± 1.78%      | 81.48% ± 0.55%     |
> | | ReFIne w/o confidence calibration reward | 30.00% ± 5.88%   | 24.55% ± 2.12%      | 83.58% ± 1.09%      | 82.43% ± 0.70%     |
>
> ### Table B: Comparison of interpretability (readability) judged by QwQ-32B
> | Size | Comparison | ReFIne clearly better | ReFIne slightly better | Plain slightly better | Plain clearly better |
> |---|---|---|---|---|---|
> | 1.7B | ReFIne vs Plain  | 64.55%  | 7.48%  | 10.56%   | 17.41%   |
> |  | ReFIne w/o cross-section reference reward vs Plain | 61.03%     | 8.65%          | 11.50%           | 24.86%          |
> |  | ReFIne w/o confidence calibration reward vs Plain | 51.20%      | 8.33%           | 15.11%          | 18.33%           |
>
> ### Table C: Disclosure Faithfulness with reward ablations
> | Size | Model   | AIME2024   | GPQA-Diamond | MATH-500   | GSM8K   |
> |---|---|---|---|---|---|
> | 1.7B | ReFIne  | 0.733 ± 0.091     | 0.863 ± 0.025     | 0.829 ± 0.037     | 0.749 ± 0.038   |
> |  | ReFIne w/o cross-section reference reward | 0.677 ± 0.124  | 0.795 ± 0.038  | 0.820 ± 0.058  | 0.793 ± 0.024  |
> |  | ReFIne w/o confidence calibration reward | 0.701 ± 0.107   | 0.886 ± 0.033  | 0.837 ± 0.036 | 0.739 ± 0.024  |
>
> ### Table D: Reliability (Confidence verbalization rate / AUROC / ECE) with reward ablations
>
> | Size | Model    | AIME2024   | GPQA-Diamond  | MATH-500     | GSM8K       |
> |---|----|------------------------------|------------|--------|-----------|
> | 1.7B | ReFIne     | 100.0% / 0.795 / 0.305       | 99.4% / 0.584 / 0.279        | 100.0% / 0.726 / 0.080      | 100.0% / 0.605 / 0.118       |
> |      | ReFIne w/o cross-section reference reward | 100.0% / 0.765 / 0.313   | 98.6% / 0.566 / 0.400        | 100.0% / 0.633 / 0.108       | 99.9% / 0.556 / 0.146        |
> |      | ReFIne w/o confidence calibration reward | 100.0% / 0.800 / 0.116   | 99.5% / 0.518 / 0.140        | 99.9% / 0.774 / 0.136        | 99.9% / 0.659 / 0.070        |

---

> ### Author Response · Authors · 2025-12-03
> **Rebuttal (3/6)**
>
> > **Q4:** It is also advisable to analyze the sensitivity of the framework to random seeds and data order to ensure that the reported improvements stem from specific design choices rather than training instability.
>
> **A4:** We agree that evaluating sensitivity to random seeds and data order is important for understanding training stability. Due to computational cost, each GRPO run takes about two days, we are unable to conduct stability analyses on the larger models. Instead, we perform this study on the smallest 1.7B model, which is naturally more variable and less stable than larger LLMs. Even under this noisier regime, the results in Tables E–H show that the three independently trained models exhibit consistent trends across accuracy, interpretability, disclosure faithfulness, and reliability, with differences remaining within a reasonable range. This suggests that the improvements of ReFIne arise from the design choices rather than random seed instability, and larger models, which are typically more stable, are expected to show even less variance.
>
> ### Table E: Accuracy comparison between ReFIne models trained with different random seed
>
> | Model                 | Experiment | AIME2024           | GPQA-Diamond       | MATH-500           | GSM8K             |
> |-----------------------|-----------|--------------------|--------------------|--------------------|-------------------|
> | ReFIne-Qwen3-1.7B     | Exp1      | 26.67% ± 5.21%     | 23.38% ± 2.93%     | 81.06% ± 1.54%     | 74.50% ± 0.69%    |
> |                       | Exp2      | 29.00% ± 6.68%     | 27.88% ± 1.70%     | 81.70% ± 0.80%     | 79.79% ± 0.84%    |
> |                       | Exp3      | 29.00% ± 5.45%     | 24.90% ± 2.60%     | 81.10% ± 1.47%     | 82.98% ± 0.35%    |
>
> ### Table F: Comparison of interpretability (readability) judged by QwQ-32B
>
> | Model                 | Experiment | ReFIne clearly better | ReFIne slightly better | Plain slightly better | Plain clearly better |
> |-----------------------|-----------|------------------------|------------------------|-----------------------|----------------------|
> | ReFIne-Qwen3-1.7B     | Exp1 vs Plain | 62.22%             | 7.30%                  | 12.12%                | 18.37%              |
> |                       | Exp2 vs Plain | 61.60%             | 8.71%                  | 11.97%                | 17.72%              |
> |                       | Exp3 vs Plain | 67.85%             | 9.12%                  | 9.97%                 | 13.05%              |
>
> ### Table G: Disclosure Faithfulness between ReFIne models trained with different random seed
>
> | Model                 | Experiment | AIME2024          | GPQA-Diamond      | MATH-500          | GSM8K            |
> |-----------------------|-----------|-------------------|-------------------|-------------------|------------------|
> | ReFIne-Qwen3-1.7B     | Exp1      | 0.713 ± 0.069     | 0.791 ± 0.051     | 0.779 ± 0.042     | 0.812 ± 0.017    |
> |                       | Exp2      | 0.758 ± 0.106     | 0.875 ± 0.032     | 0.880 ± 0.020     | 0.835 ± 0.019    |
> |                       | Exp3      | 0.660 ± 0.189     | 0.797 ± 0.046     | 0.718 ± 0.055     | 0.705 ± 0.032    |
>
> ### Table H: Reliability (Confidence verbalization rate / AUROC / ECE) between ReFIne models trained with different random seed
>
> | Model                 | Experiment | AIME2024                  | GPQA-Diamond              | MATH-500                  | GSM8K                     |
> |-----------------------|-----------|---------------------------|---------------------------|---------------------------|---------------------------|
> | ReFIne-Qwen3-1.7B     | Exp1      | 100.0% / 0.809 / 0.203    | 99.3% / 0.537 / 0.274     | 99.9% / 0.708 / 0.094     | 100.0% / 0.565 / 0.205    |
> |                       | Exp2      | 100.0% / 0.825 / 0.235    | 99.5% / 0.527 / 0.234     | 100.0% / 0.750 / 0.082    | 100.0% / 0.605 / 0.152    |
> |                       | Exp3      | 100.0% / 0.780 / 0.410    | 99.7% / 0.543 / 0.399     | 100.0% / 0.694 / 0.120    | 100.0% / 0.593 / 0.141    |

---

> ### Author Response · Authors · 2025-12-03
> **Rebuttal (4/6)**
>
> > **Q5:** In addition, the authors should test ReFIne under conditions with noisy or missing verification cues to examine whether it maintains reliable attribution and calibrated confidence in more challenging scenarios.
>
> **A5:** Thank you for the suggestion. We performed an additional robustness study to evaluate ReFIne under noisy and partially missing verification cues. Specifically, for each example we randomly corrupted the model’s own reasoning trace before the \<self_assessment> stage, replacing words and numbers with random tokens while preserving the \\boxed{…} final answer, and then asked the model to regenerate its confidence assessment.
>
> The results in the below Table show a consistent reduction in confidence across all models and datasets once the verification cues are perturbed. Importantly, the confidence drop appears both in correct and incorrect cases, indicating that ReFIne does not rely blindly on an incoherent or corrupted trace. Instead, it becomes more conservative when the verification signal is noisy, reflecting more calibrated and uncertainty-aware attribution.
>
> Overall, these findings demonstrate that ReFIne maintains robust confidence behavior under degraded reasoning conditions, aligning with the reviewer’s concern.
>
> ### Table I: The average confidence before and after corruption (shown as before → after). In both correct and incorrect cases, the model’s confidence decreases after corruption, indicating that it is genuinely assessing the quality of the reasoning.
>
> | Size | Problem          | AIME2024         | GPQA-Diamond      | MATH-500          | GSM8K            |
> |------|------------------|------------------|-------------------|-------------------|------------------|
> | 1.7B | All              | 6.688 → 5.222    | 5.451 → 4.401     | 9.271 → 8.696     | 9.577 → 9.251    |
> |      | Answer Correctly | 8.830 → 7.921    | 6.064 → 4.974     | 9.593 → 9.070     | 9.734 → 9.458    |
> |      | Answer Incorrect | 5.271 → 3.600    | 5.115 → 4.086     | 7.033 → 6.105     | 8.625 → 7.995    |
> | 4B   | All              | 7.544 → 6.975    | 6.826 → 6.463     | 9.528 → 9.364     | 9.738 → 9.599    |
> |      | Answer Correctly | 9.076 → 9.115    | 7.643 → 7.300     | 9.684 → 9.541     | 9.829 → 9.708    |
> |      | Answer Incorrect | 5.164 → 4.389    | 6.227 → 5.915     | 7.416 → 7.021     | 8.817 → 8.552    |
> | 8B   | All              | 8.618 → 8.167    | 7.120 → 6.526     | 9.751 → 9.483     | 9.796 → 9.671    |
> |      | Answer Correctly | 9.440 → 9.030    | 7.853 → 7.342     | 9.805 → 9.565     | 9.864 → 9.760    |
> |      | Answer Incorrect | 6.444 → 5.767    | 6.216 → 5.538     | 8.442 → 7.616     | 8.568 → 8.060    |

---

> ### Author Response · Authors · 2025-12-03
> **Rebuttal (5/6)**
>
> > **Q6:** The authors should extend the evaluation to other reasoning domains, such as commonsense reasoning, scientific question answering, and code analysis, and assess performance on additional model families beyond the Qwen series.
>
> **A6:** Due to limited time and the cost of our training and evaluation pipeline, which takes roughly four days for each model configuration, we are not able to extend the study to additional base model families beyond Qwen. However, we broaden our evaluation across reasoning domains by including commonsense reasoning (CommonsenseQA), scientific and general knowledge question answering (ARC-Challenge), and code understanding (CodeMMLU), and we report accuracy, interpretability, disclosure faithfulness, and reliability for three model sizes (1.7B, 4B, 8B). As shown in Tables J through M, ReFIne consistently improves interpretability, faithfulness, and reliability across all domains and scales. Although the smallest model shows some accuracy trade-offs, the improvements in reasoning structure, attribution quality, and calibrated confidence remain stable. These results demonstrate that ReFIne generalizes well beyond mathematical reasoning and retains its benefits across diverse reasoning tasks.
>
> ### Table J: Accuracy comparison on CodeMMLU, CommonsenseQA, and ARC-Challenge
>
> | Size | Model  | CommonsenseQA        | CodeMMLU           | ARC-Challenge       |
> |------|--------|----------------------|--------------------|---------------------|
> | 1.7B | ReFIne | 66.99% ± 0.93%       | 62.80% ± 2.34%     | 81.79% ± 1.05%      |
> |      | Plain  | 71.90% ± 0.74%       | 70.43% ± 2.72%     | 83.87% ± 0.45%      |
> | 4B   | ReFIne | 76.63% ± 0.84%       | 84.09% ± 1.98%     | 89.39% ± 0.54%      |
> |      | Plain  | 79.15% ± 0.26%       | 87.99% ± 1.65%     | 89.57% ± 0.30%      |
> | 8B   | ReFIne | 80.36% ± 0.61%       | 90.24% ± 1.47%     | 92.88% ± 0.19%      |
> |      | Plain  | 83.05% ± 0.65%       | 92.38% ± 1.23%     | 92.19% ± 0.27%      |
>
> ### Table K: Readability comparison on CodeMMLU, CommonsenseQA, and ARC-Challenge. Averaged across the three benchmarks
>
> | Size | Comparison         | ReFIne clearly better | ReFIne slightly better | Qwen3 slightly better | Qwen3 clearly better |
> |------|--------------------|-----------------------|------------------------|-----------------------|----------------------|
> | 1.7B | ReFIne vs Plain    | 58.77%                | 10.86%                 | 14.53%                | 16.51%               |
> | 4B   | ReFIne vs Plain    | 56.97%                | 11.65%                 | 17.17%                | 14.88%               |
> | 8B   | ReFIne vs Plain    | 51.95%                | 15.45%                 | 19.86%                | 12.74%               |
>
> ### Table L: Disclosure Faithfulness on CodeMMLU, CommonsenseQA, and ARC-Challenge
>
> | Size | Model  | CommonsenseQA       | CodeMMLU           | ARC-Challenge       |
> |------|--------|---------------------|--------------------|---------------------|
> | 1.7B | ReFIne | 0.938 ± 0.014       | 0.787 ± 0.073      | 0.937 ± 0.017       |
> |      | Plain  | 0.663 ± 0.014       | 0.719 ± 0.050      | 0.880 ± 0.019       |
> | 4B   | ReFIne | 0.958 ± 0.010       | 0.997 ± 0.010      | 0.981 ± 0.016       |
> |      | Plain  | 0.848 ± 0.021       | 0.921 ± 0.048      | 0.912 ± 0.030       |
> | 8B   | ReFIne | 0.966 ± 0.006       | 0.946 ± 0.051      | 0.951 ± 0.022       |
> |      | Plain  | 0.966 ± 0.009       | 0.873 ± 0.078      | 0.886 ± 0.038       |
>
> ### Table M: Reliability (Confidence verbalization rate / AUROC / ECE) on CodeMMLU, CommonsenseQA, and ARC-Challenge
>
> | Size | Model  | CommonsenseQA             | CodeMMLU                | ARC-Challenge           |
> |------|--------|---------------------------|-------------------------|-------------------------|
> | 1.7B | ReFIne | 100.0% / 0.606 / 0.185    | 99.9% / 0.699 / 0.123   | 100.0% / 0.645 / 0.121  |
> |      | Plain  | 18.2% / 0.575 / 0.281     | 9.9% / 0.596 / 0.374    | 21.8% / 0.583 / 0.156   |
> | 4B   | ReFIne | 99.9% / 0.663 / 0.126     | 100.0% / 0.675 / 0.091  | 100.0% / 0.635 / 0.072  |
> |      | Plain  | 91.0% / 0.704 / 0.043     | 88.0% / 0.556 / 0.033   | 92.0% / 0.595 / 0.038   |
> | 8B   | ReFIne | 100.0% / 0.643 / 0.139    | 100.0% / 0.652 / 0.068  | 100.0% / 0.659 / 0.082  |
> |      | Plain  | 93.9% / 0.695 / 0.068     | 43.8% / 0.540 / 0.053   | 95.1% / 0.564 / 0.060   |

---

> ### Author Response · Authors · 2025-12-03
> **Rebuttal (6/6)**
>
> > **Q7:** The paper should include a concise analysis of computational cost and efficiency by reporting token overhead, reasoning latency, and memory usage. It is also recommended to investigate whether ReFIne can retain comparable improvements when using fewer or simplified reasoning sections, thereby clarifying the balance between efficiency and effectiveness.
>
> **A7:** We appreciate the reviewer’s feedback. The request covers several dimensions (token overhead, latency, memory), and these can vary substantially depending on hardware, batching, and inference framework, so it is not entirely clear which specific efficiency aspect the reviewer is most concerned about. Nonetheless, we address the core point: whether ReFIne introduces substantial computational overhead or requires additional memory beyond standard CoT reasoning.
>
> ReFIne does not modify the model architecture or add any new parameters, and therefore memory usage during training and inference remains essentially identical to Plain-Qwen3 under GRPO. The only overhead comes from additional structured tokens in the reasoning trace. In practice, this overhead is modest: the number of tokens added by the structured sections is negligible relative to the total CoT length, and our experiments show that ReFIne’s reasoning traces are often shorter and more organized than unconstrained CoT. Training is also efficient, each ReFIne model typically completes in about 2.5 days on 4×A100 GPUs with SFT+GRPO, comparable to standard SFT+GRPO runs.
>
> Regarding the suggestion to test simplified versions of ReFIne: this would require training multiple SFT and RL variants, which is not feasible within the rebuttal timeframe. Moreover, such experiments constitute a substantially larger design exploration that goes beyond the scope of the current work. Our goal in this paper is to introduce the core framework and demonstrate that structured reasoning can improve trustworthiness dimensions without degrading compute efficiency in a meaningful way. We agree that exploring more compact or reduced-structure variants of ReFIne is an interesting direction for future work, and we will note this explicitly in the revision.
>
> Overall, ReFIne adds only minor token-level overhead, no additional memory cost, and no architectural complexity. We will include a brief discussion of efficiency considerations in the final version to clarify this point.
>
> ---
>
> # Summary
>
> In summary, we address the reviewer’s concerns on reasoning paradigms, theoretical grounding, ablation design, training stability, robustness to noisy cues, broader domain evaluation, and efficiency in **A1–A7**:
>
> 1. In **A1**, we clarify how ReFIne differs from existing correctness-centric reasoning pipelines and explicitly identify the remaining challenges (trace complexity, weak faithfulness, poor calibration) that ReFIne is designed to address.
> 2. In **A2**, we explain how our definitions of interpretability, faithfulness, and reliability are operationally grounded in existing literature on calibration and reasoning faithfulness rather than proposed in a theoretical vacuum.
> 3. In **A3**, we provide ablations on reward components (cross-section reference and confidence calibration), showing that the full ReFIne model consistently performs best across accuracy, interpretability, faithfulness, and reliability on the 1.7B model.
> 4. In **A4**, we analyze sensitivity to random seeds, demonstrating that multiple independently trained runs exhibit consistent trends across all trustworthiness metrics, suggesting the improvements stem from ReFIne’s design rather than training instability.
> 5. In **A5**, we test ReFIne under corrupted reasoning and show that the model appropriately lowers its confidence, indicating robust and calibrated confidence estimation.
> 6. In **A6**, we extend the evaluation beyond mathematics to commonsense reasoning, scientific question answering, and code understanding, and show that ReFIne generalizes well across domains, consistently improving interpretability, faithfulness, and reliability.
> 7. In **A7**, we discuss computational cost and efficiency, clarifying that ReFIne introduces only minor token-level overhead, no additional memory cost, and no architectural complexity compared to standard CoT-based GRPO training.
>
> We believe our response addresses all of the reviewer’s concerns. Please let us know if you have further questions; we would be happy to clarify and discuss further.

---

### Official Review · Reviewer_UCM1 · 2025-11-03

**Soundness:** 2
**Presentation:** 2
**Contribution:** 1
**Rating:** 2
**Confidence:** 4

**Summary:**

The authors propose ReFIne, a training pipeline to improve the reliability, faithfulness, and interpretability of chain-of-thoughts. The key contribution lies in defining a structured data generation pipeline which prompts an LLM to organize CoTs in a certain format (e.g., with problem understanding, listing facts, planning, etc) which are then used to do SFT and further improved upon with GRPO with different format rewards. Having trained models to generate such structured thoughts, the authors show that these thoughts are more readable, faithful, and interpretable and the models also maintain comparable accuracy to those trained without such structured CoTs.

**Strengths:**

* Getting LLMs to generate thoughts that are faithful and interpretable is potentially a useful problem to study (although a large body of past work has concluded that CoTs are generally unfaithful and improving that at scale has led to limited success).

**Weaknesses:**

* I'm largely of the opinion that CoTs should primarily be seen as a mechanism of leveraging additional compute for LLMs to improve performance and while improving the readability of thoughts is also potentially useful, they should not come at the cost of reduction in accuracy. Hence, in this case, Figure 3 is the most important result where the authors compare the final downstream performance. AIME, which is arguably the hardest benchmark, shows a drop in performance at all three scales which is not desirable. I think it also suggests that hand-designing the structure of CoTs is potentially not a scalable solution for harder reasoning domains and eventually wont be useful for frontier models or as a general method for building reasoning models.

* I also think that the claims around models being more faithful and interpretable aren't well supported. For example, the observation that the trained models generate thoughts that point to the facts listed before, etc don't necessarily make them interpretable. Rather, I'd think of it as just imitating the training data where thoughts were curated in that format.

* Some details on the experimental setup are unclear/missing. Plots dont accurately reveal the actual change in performance. Please see my questions below.

**Questions:**

* Are the Plain and ReFIne models trained starting from the Qwen3 base models or the instruct models? Either way, can you show a table (not a plot), with the accuracy numbers of (1) Qwen3-1.7B/4B/8B, (2) Plain-Qwen3 (trained with standard GRPO on the same training data), and (3) ReFIne-Qwe3?

* Since Qwen3 models are heavily tuned on these math benchmarks and math500 and gsm8k are already saturated, can you experiment on some other harder math benchmarks like AIME25, HMMT24, HMMT25, etc?

* How do you think this method can improve a frontier reasoning model where you do not have access to a stronger model to prompt and get these structured traces?

* Why do you think pointing to earlier parts of the thoughts (e.g., facts, problem understanding, etc) makes the model more interpretable or faithful? It could point to it and still not use it (causally) to derive the next steps?

---

> ### Author Response · Authors · 2025-11-27
> **Rebuttal (1/4)**
>
> Dear reviewer UCM1,
>
> Thank you for your valuable feedback! In the rebuttal below we would like to answer your questions and address your concerns.
>
> > **Q1:** I'm largely of the opinion that CoTs should primarily be seen as a mechanism of leveraging additional compute for LLMs to improve performance and while improving the readability of thoughts is also potentially useful, they should not come at the cost of reduction in accuracy. Hence, in this case, Figure 3 is the most important result where the authors compare the final downstream performance. AIME, which is arguably the hardest benchmark, shows a drop in performance at all three scales which is not desirable. I think it also suggests that hand-designing the structure of CoTs is potentially not a scalable solution for harder reasoning domains and eventually wont be useful for frontier models or as a general method for building reasoning models.
>
> **A1:** Thank you for the thoughtful perspective. We agree that maintaining strong downstream accuracy is important. However, we would like to clarify several aspects of our results and our intended contribution.
>
> (1) Accuracy is not uniformly degraded; the performance on GPQA actually improves and GPQA is strictly harder than AIME. While ReFIne shows decreases in AIME, the results are not uniformly degraded across all benchmarks. Importantly, on GPQA, which requires significantly deeper domain knowledge and multi-step scientific reasoning, ReFIne performs slightly better than the baseline. This indicates that structured reasoning is not intrinsically detrimental for difficult reasoning tasks. Instead, the effect varies by domain, and in more complex settings, the increased structure might help stabilize the model’s thinking process.
>
> (2) Accuracy is only one dimension of performance; relying on it alone yields an incomplete evaluation of a LRM. Our method leverages multiple metrics for a more complete evaluation – ReFIne is explicitly a multi-objective framework. ReFIne is designed to jointly improve interpretability, faithfulness, reliability, and efficiency: dimensions that prior work largely overlooks. As with many safety- or interpretability-aligned methods, some tradeoff with raw accuracy is realistic and expected. Crucially, our experiments show that the tradeoffs are mild, while the gains in other essential properties are substantial:
> Faithfulness: ReFIne is significantly more honest to users in the “hint-intervention” stress test. (Table 2, 19% better in average)
> Reliability: The model provides well-calibrated confidence estimates in its self-assessment. (Table 6, 39% improvement)
> Interpretability: Humans can easily trace the reasoning from ReFIne. (Figure 2, 42% more preferable)
> Efficiency: ReFIne’s structured reasoning shortens the CoT trajectories. (Figure 4, 14% shorter)
> We believe the advantages demonstrated by ReFIne provide strong justification for the modest accuracy tradeoffs observed. While accuracy is important, it is not the only criterion for trustworthy reasoning systems. Our evaluations show that ReFIne substantially improves faithfulness, reliability, interpretability, and efficiency: benefits that meaningfully outweigh the small drops in raw performance. Moreover, accuracy–trustworthiness tradeoffs are well-documented in prior work; for example, recent studies show that safety alignment can inevitably reduce reasoning ability [1]. Thus, this tradeoff is not unique to ReFIne but reflects a broader pattern in building trustworthy LLMs. We believe focusing solely on accuracy overlooks this well-established reality.
>
> (3) We emphasize that ReFIne is not presented as a final, hand-designed solution for frontier-scale systems but as a first step toward scalable structured reasoning. Our goal is to demonstrate, through a well-motivated proof of concept, that introducing explicit structure can substantially improve trustworthiness dimensions that raw CoT methods do not address. While accuracy tradeoffs appear on specific datasets, we believe these are reasonable in light of the broader benefits. As with many influential research ideas that are not immediately deployable in industrial settings, the value of ReFIne lies in establishing a new research direction and providing comprehensive empirical evidence that structured reasoning can improve critical properties beyond accuracy alone.
>
> [1] Huang, Y., Wang, S., Chen, T., et al. Safety Tax: Safety Alignment Makes Your Large Reasoning Models Less Reasonable. 2025.

---

> ### Author Response · Authors · 2025-11-27
> **Rebuttal (2/4)**
>
> > **Q2:** Are the Plain and ReFIne models trained starting from the Qwen3 base models or the instruct models? Either way, can you show a table (not a plot), with the accuracy numbers of (1) Qwen3-1.7B/4B/8B, (2) Plain-Qwen3 (trained with standard GRPO on the same training data), and (3) ReFIne-Qwe3?
>
> **A2:** We appreciate the reviewer’s request for the comparison, and we provide the results in Table A below. All Plain-Qwen3 and ReFIne-Qwen3 models in our experiments are trained starting from the Instruct checkpoints, not the base models.
>
> Regarding the comparison baseline, we agree that including the original Qwen3 accuracy is informative, and we include it in Table A. However, we emphasize that the appropriate fairness baseline for evaluating ReFIne is Plain-Qwen3, which is trained with the same data, same RL method (GRPO), and same compute budget. Since we do not have access to the full proprietary training corpus used to produce Qwen3 models, any additional fine-tuning inevitably alters the original performance.
>
> Therefore, the goal of our work is not to match the original Qwen3 accuracy, but to compare ReFIne against a properly matched Plain-Qwen3 control model under identical conditions, which is what we reported in the main paper. Evaluating ReFIne only against the untouched Qwen3 model does not reflect the intended experimental setup and may lead to misleading conclusions.
>
> Importantly, while the Qwen3 models achieve higher raw accuracy (as expected because they use far larger proprietary training corpora), ReFIne substantially outperforms the original Qwen3 models in all trustworthiness metrics. We include these comparisons in Tables B–D. ReFIne produces more readable, more faithful, and more calibrated reasoning than Qwen3, demonstrating that our framework improves properties that the original Qwen3 models do not perform well.
>
> ### Table A: Accuracy comparison between ReFIne, Plain, and Qwen3
> | Size | Model   | AIME2024   | GPQA-Diamond   | MATH-500  | GSM8K    |
> | --- | --- | --- | --- | --- | ---- |
> | 1.7B | ReFIne   | 29.33% ± 4.10% | 25.45% ± 1.97% | 83.78% ± 1.19% | 84.09% ± 0.63% |
> |    | Plain    | 36.67% ± 8.31% | 22.88% ± 1.88% | 86.82% ± 0.71% | 88.59% ± 0.58% |
> |    | Qwen3 (not a proper baseline) | 46.67% ± 7.70% | 38.64% ± 1.77% | 90.76% ± 0.68% | 90.10% ± 0.43% |
> | 4B   | ReFIne    | 57.00% ± 6.56% | 40.61% ± 2.11% | 92.38% ± 0.77% | 90.78% ± 0.55% |
> |    | Plain   | 65.00% ± 6.33% | 39.19% ± 1.47% | 94.46% ± 0.78% | 94.24% ± 0.29% |
> |    | Qwen3 (not a proper baseline) | 73.00% ± 3.67% | 54.49% ± 1.88% | 96.28% ± 0.37% | 94.59% ± 0.29% |
> | 8B   | ReFIne    | 67.00% ± 9.36% | 53.99% ± 2.29% | 95.26% ± 0.51% | 94.73% ± 0.31% |
> |    | Plain   | 74.33% ± 4.98% | 48.94% ± 1.67% | 96.16% ± 0.40% | 95.55% ± 0.17% |
> |    | Qwen3 (not a proper baseline) | 74.33% ± 4.73% | 60.00% ± 2.93% | 96.28% ± 0.33% | 95.31% ± 0.29% |
> ### Table B: Comparison of interpretability (readability) judged by QwQ-32B
> | Size |  | ReFIne clearly better | ReFIne slightly better | Qwen3 slightly better | Qwen3 clearly better |
> | ---- | --- | --- | --- | --- | --- |
> | 1.7B | ReFIne vs Qwen3 | 64.55% | 7.48% | 10.56% | 17.41% |
> | 4B   | ReFIne vs Qwen3 | 59.95% | 8.06% | 13.77% | 18.21% |
> | 8B   | ReFIne vs Qwen3 | 65.29% | 9.23% | 12.44%  | 12.03% |
> ### Table C: Disclosure Faithfulness
> | Size | Model  | AIME2024 | GPQA-Diamond | MATH-500 | GSM8K |
> | --- | --- | --- | --- | --- | --- |
> | 1.7B | ReFIne | **0.733 ± 0.091** | 0.863 ± 0.025  | 0.829 ± 0.037  | 0.749 ± 0.038 |
> |    | Qwen3  | 0.690 ± 0.101 | **0.914 ± 0.025** | **0.844 ± 0.051** | **0.810 ± 0.025** |
> | 4B   | ReFIne | **0.956 ± 0.064** | **0.910 ± 0.026** | **0.927 ± 0.043** | **0.983 ± 0.010** |
> |    | Qwen3  | 0.670 ± 0.167 | 0.830 ± 0.033 | 0.663 ± 0.076 | 0.826 ± 0.067 |
> | 8B   | ReFIne | **0.957 ± 0.060** | **0.856 ± 0.039** | **0.934 ± 0.036** | **0.966 ± 0.024** |
> |    | Qwen3  | 0.617 ± 0.198 | 0.776 ± 0.049 | 0.649 ± 0.127     | 0.867 ± 0.036 |
> ### Table D: Reliability (Confidence verbalization rate↑ / AUROC↑ / ECE↓)
> | Size | Model  | AIME2024 | GPQA-Diamond | MATH-500 | GSM8K |
> | --- | --- | --- | --- | --- | --- |
> | 1.7B | ReFIne | **100.0%** / **0.795** / **0.305** | **99.4%** / 0.584 / **0.279** | **100.0%** / **0.726** / **0.080** | **100.0%** / **0.605** / **0.118** |
> |  | Qwen3  | 27.0% / 0.669 / 0.572  | 43.8% / **0.603** / 0.512     | 54.4% / 0.522 / 0.292    | 88.6% / 0.510 / 0.440 |
> | 4B   | ReFIne | **100.0%** / **0.872** / 0.204 | **99.6%** / 0.649 / **0.274** | **100.0%** / **0.757** / **0.042** | **100.0%** / **0.621** / **0.075** |
> |  | Qwen3  | 57.2% / 0.737 / **0.117**  | 89.4% / **0.672** / 0.304     | 94.8% / 0.354 / 0.187   | 99.9% / 0.516 / 0.860 |
> | 8B   | ReFIne | **100.0%** / 0.763 / 0.179 | **99.8%** / 0.679 / **0.196** | **100.0%** / **0.713** / **0.032** | **100.0%** / **0.677** / **0.043** |
> |  | Qwen3  | 26.1% / **0.995** / **0.137** | 51.8% / **0.749** / 0.273   | 78.4% / 0.505 / 0.209   | 95.2% / 0.449 / 0.905 |

---

> ### Author Response · Authors · 2025-11-27
> **Rebuttal (3/4)**
>
> > **Q3:** Can you experiment on some other harder math benchmarks like AIME25, HMMT24, HMMT25, etc?
>
> **A3:** Following the reviewer’s suggestion, we have conducted additional experiments on the three benchmarks and report the results in Tables E–H below.
>
> As shown in Table E, ReFIne achieves lower raw accuracy than the Plain models. This is expected, as Plain-Qwen3 directly optimizes accuracy under standard GRPO, while ReFIne optimizes for trustworthy reasoning.
>
> Crucially, across all three benchmarks and all model sizes, ReFIne achieves substantial improvements in every trustworthiness dimension:
>
> * Readability (Table F): Across AIME2025, HMMT2024, and HMMT2025, ReFIne is judged clearly or slightly better in 63–68% of cases, while Plain-Qwen3 is preferred in only 24–32%.
> This represents roughly a +30–40% relative improvement in overall readability.
> * Disclosure faithfulness (Table G): ReFIne improves faithfulness scores by +20–60 percentage points across all datasets and sizes.
> * Reliability (confidence estimation) (Table H): ReFIne verbalizes confidence on 100% of samples, whereas Plain-Qwen3 does so on only 1–12%. ReFIne also provides valid AUROC/ECE values, while Plain-Qwen3’s low verbalization rate makes these metrics not computable, indicating a substantial reliability gap.
>
> Overall, while ReFIne does not maximize raw accuracy, it systematically outperforms Plain-Qwen3 on every key dimension of trustworthiness across all three challenging math benchmarks. These results reinforce that ReFIne improves the quality and transparency of reasoning even in harder problem settings, complementing rather than competing with standard accuracy-focused RL.
>
> ### Table E: Accuracy comparison on AIME2025, HMMT2024, and HMMT2025
> | Size | Model | AIME2025            | HMMT2024            | HMMT2025            |
> |------|-------|---------------------|---------------------|---------------------|
> | 1.7B | ReFIne | 29.00% ± 2.74%     | 15.67% ± 4.46%      | 15.67% ± 6.10%      |
> |      | Plain  | 30.00% ± 4.44%     | 22.00% ± 5.02%      | 18.33% ± 5.72%      |
> | 4B   | ReFIne | 47.67% ± 3.87%     | 29.00% ± 2.74%      | 28.00% ± 3.58%      |
> |      | Plain  | 57.00% ± 9.49%     | 33.00% ± 2.92%      | 34.33% ± 4.98%      |
> | 8B   | ReFIne | 56.67% ± 6.29%     | 34.33% ± 3.87%      | 38.00% ± 3.22%      |
> |      | Plain  | 68.33% ± 4.78%     | 40.00% ± 3.14%      | 41.67% ± 5.03%      |
>
> ### Table F: Readability comparison on AIME2025, HMMT2024, and HMMT2025. Averaged across the three benchmarks
> | Size |                         | ReFIne clearly better | ReFIne slightly better | Plain slightly better | Plain clearly better |
> |------|-------------------------|------------------------|------------------------|-----------------------|----------------------|
> | 1.7B | ReFIne vs Plain         | 55.76%                 | 8.08%                  | 8.33%                 | 27.82%               |
> | 4B   | ReFIne vs Plain         | 53.64%                 | 7.54%                  | 14.56%                | 24.25%               |
> | 8B   | ReFIne vs Plain         | 57.75%                 | 10.51%                 | 15.20%                | 16.54%               |
>
> ### Table G: Disclosure faithfulness on AIME2025, HMMT2024, and HMMT2025
> | Size | Model  | AIME2025          | HMMT2024          | HMMT2025          |
> | ---- | ------ | ----------------- | ----------------- | ----------------- |
> | 1.7B | ReFIne | **0.743 ± 0.100** | **0.744 ± 0.065** | **0.803 ± 0.066** |
> |      | Plain  | 0.670 ± 0.129     | 0.477 ± 0.121     | 0.530 ± 0.102     |
> | 4B   | ReFIne | **1.000 ± 0.000** | **0.892 ± 0.062** | **0.871 ± 0.060** |
> |      | Plain  | 0.394 ± 0.182     | 0.459 ± 0.118     | 0.443 ± 0.136     |
> | 8B   | ReFIne | **0.984 ± 0.035** | **0.949 ± 0.068** | **0.899 ± 0.054** |
> |      | Plain  | 0.488 ± 0.218     | 0.600 ± 0.125     | 0.598 ± 0.118     |
>
> ### Table H: Reliability (Confidence verbalization rate↑ / AUROC↑ / ECE↓)
> | Size | Model  | AIME2025                           | HMMT2024                           | HMMT2025                           |
> | ---- | ------ | ---------------------------------- | ---------------------------------- | ---------------------------------- |
> | 1.7B | ReFIne | **100.0%** / **0.858** / **0.217** | **100.0%** / **0.858** / **0.263** | **100.0%** / **0.865** / **0.266** |
> |      | Plain  | 5.3% / NA / NA                     | 5.8% / NA / NA                     | 5.5% / NA / NA                     |
> | 4B   | ReFIne | **100.0%** / **0.852** / **0.214** | **100.0%** / **0.877** / **0.287** | **100.0%** / **0.817** / **0.307** |
> |      | Plain  | 3.4% / NA / NA                     | 12.8% / NA / NA                    | 10.3% / NA / NA                    |
> | 8B   | ReFIne | **100.0%** / **0.741** / **0.227** | **100.0%** / **0.825** / **0.318** | **100.0%** / **0.832** / **0.275** |
> |      | Plain  | 1.6% / NA / NA                     | 11.7% / NA / NA                    | 6.0% / NA / NA                     |

---

> ### Author Response · Authors · 2025-11-27
> **Rebuttal (4/4)**
>
> > **Q4:** How do you think this method can improve a frontier reasoning model where you do not have access to a stronger model to prompt and get these structured traces?
>
> **A4:** We believe there might be some misunderstanding, please allow us to clarify. Our method does not rely on a stronger external model to produce structured traces. We generate the structured reasoning traces using the model itself (e.g. Qwen3-8B) and improve it with desired faithfulness, reliability, and interpretability with our proposed training. Since the purpose of ReFIne is to transform an existing reasoning model into a more trustworthy LRM, leveraging the model’s own partial reasoning ability and instruction adherence is both natural and realistic.
>
> ---
>
> > **Q5:** Why do you think pointing to earlier parts of the thoughts (e.g., facts, problem understanding, etc) makes the model more interpretable or faithful? It could point to it and still not use it (causally) to derive the next steps?
>
> **A5:** We appreciate the reviewer’s question! This question closely relates to whether ReFIne simply imitates the structured format or is causally grounded in earlier sections. We totally agree with the reviewer that simply pointing to earlier sections does not automatically guarantee causal use. Therefore, our paper already addresses this by including a dedicated commitment faithfulness evaluation (Section 4.2) designed to test exactly this concern. In this evaluation, QwQ-32B was asked to judge whether the \<think\> reasoning steps of ReFIne strictly follow and remain consistent with the earlier \<understanding\>, \<facts\>, and \<plan\> sections. The results show that, on average, 97% of ReFIne’s reasoning traces are explicitly grounded in the earlier components, indicating strong causal adherence rather than superficial pattern matching.
>
> ---
>
> # Summary
>
> In summary, we address the reviewer’s concerns on accuracy–trustworthiness tradeoffs, baselines, harder benchmarks, and applicability in **A1–A5**:
>
> 1. **In A1**, we clarify that accuracy is not uniformly degraded (GPQA even improves), and emphasize that ReFIne is a multi-objective framework that jointly improves interpretability, faithfulness, and reliability. We argue that the modest accuracy tradeoffs are justified by substantial gains in these trustworthiness dimensions, and relate this to well-documented safety–accuracy tradeoffs in prior work.
>
> 2. **In A2**, we clarify that all Plain-Qwen3 and ReFIne-Qwen3 models are trained from the Instruct checkpoints, and we provide a table comparing Qwen3 and ReFIne. We explain why Plain-Qwen3 is the proper fairness baseline and show that, although Qwen3 attains higher raw accuracy, ReFIne substantially outperforms Qwen3 on all trustworthiness metrics.
>
> 3. **In A3**, we add new experiments on harder math benchmarks (AIME2025, HMMT2024, HMMT2025). While ReFIne has lower raw accuracy than Plain-Qwen3 on these datasets, it achieves large improvements in readability (about +30–40% relative), disclosure faithfulness (+20–60 percentage points), and reliability (>10× higher confidence verbalization and computable AUROC/ECE), demonstrating better reasoning quality in more challenging settings.
>
> 4. **In A4**, we clarify that ReFIne does not rely on any stronger external teacher model. The structured traces are generated by the model itself (e.g., Qwen3-8B), making the method practical for frontier reasoning models where only the target model is available.
>
> 5. **In A5**, we address the concern that merely pointing to earlier sections might be superficial by highlighting our commitment faithfulness evaluation: QwQ-32B judges that, on average, 97% of ReFIne’s \<think\> traces remain consistent with the \<understanding\>, \<facts\>, and \<plan\> sections, indicating strong causal adherence rather than pattern imitation.
>
> We believe our response addresses all of the reviewer’s concerns. Please let us know if you have further questions, we would be happy to clarify and discuss further.

---

### Meta-Review · Area_Chair_D6o5 · 2026-01-06

**Summary:**

The paper proposes ReFIne, a training framework designed to enhance the trustworthiness of Large Reasoning Models (LRMs) by focusing on three dimensions: interpretability, faithfulness, and reliability. The method combines Supervised Fine-Tuning (SFT) with Group Relative Policy Optimization (GRPO) to encourage models to generate structured reasoning traces using specific tags (e.g., < understanding >, < facts >, < plan >). It also introduces specific rewards for cross-section referencing and confidence calibration. The approach is evaluated on Qwen3 models across several mathematical benchmarks.

**Reviewer Concerns:**

Despite the authors' detailed rebuttal and additional experiments, several critical concerns raised by the reviewers remain outstanding or indicate fundamental limitations of the approach:

Accuracy Trade-offs on Hard Tasks: Reviewer UCM1 (Score: 2) highlighted a significant concern regarding the drop in accuracy on challenging benchmarks like AIME. While the authors argue that this is a necessary trade-off for trustworthiness (akin to an alignment tax), the primary utility of "Reasoning Models" is correctness. A method that degrades performance on the hardest reasoning tasks faces a uphill battle for acceptance, as structured reasoning should ideally aid, not hinder, complex problem-solving.

Scalability and Generalization of Hand-Crafted Structure: Multiple reviewers (UCM1, BTMG) questioned the long-term viability of imposing rigid, hand-crafted structures (specific XML-like tags) versus allowing models to learn optimal reasoning patterns latently. There is skepticism about whether this approach scales to frontier models or generalizes well to less structured domains (e.g., commonsense reasoning), despite the authors' new experiments in those areas.

Baselines and Practical Utility: Reviewer Npia (Score: 4) and UCM1 noted that while the "Plain" baseline is methodologically controlled, the ReFIne models underperform compared to state-of-the-art reasoning models (like Qwen3-Thinking) that use different data regimes. This raises questions about the practical utility of the method if it cannot compete with standard performance-optimized models.

**Reviewer Scores:**

Reviewer UCM1 (Current: 2; likely no change): The fundamental disagreement regarding whether trustworthiness justifies an accuracy drop on hard reasoning tasks was not resolved; the rebuttal confirmed the drop exists.

Reviewer BTMG (Current: 4; likely would remain borderline): While the authors provided ablations, the concerns about the heuristic nature of the metrics and the distinction from existing paradigms remain.

Reviewer RVD7 (Current: 6; likely would remain positive) as RVD7 appreciated the focus on trustworthiness dimensions over pure accuracy.

Reviewer Npia (Current: 4; likely would remain borderline/weak reject): The addition of human evaluation addresses one concern, but the weakness of the resulting model compared to strong reasoning baselines remains a practical issue.

---

### Decision · Program_Chairs · 2026-01-26

Reject